# RNA-programmed genome editing in human cells

**Martin Jinek[1,2], Alexandra East[2], Aaron Cheng[2], Steven Lin[1,2], Enbo Ma[2], Jennifer Doudna[1,2,3,4]\***

[1]Howard Hughes Medical Institute, University of California, Berkeley, Berkeley, United States; [2]Department of Molecular and Cell Biology, University of California, Berkeley, Berkeley, United States; [3]Department of Chemistry, University of California, Berkeley, Berkeley, United States; [4]Physical Biosciences Division, Lawrence Berkeley National Laboratory, Berkeley, United States

**Abstract** Type II CRISPR immune systems in bacteria use a dual RNA-guided DNA endonuclease, Cas9, to cleave foreign DNA at specific sites. We show here that Cas9 assembles with hybrid guide RNAs in human cells and can induce the formation of double-strand DNA breaks (DSBs) at a site complementary to the guide RNA sequence in genomic DNA. This cleavage activity requires both Cas9 and the complementary binding of the guide RNA. Experiments using extracts from transfected cells show that RNA expression and/or assembly into Cas9 is the limiting factor for Cas9-mediated DNA cleavage. In addition, we find that extension of the RNA sequence at the 3′ end enhances DNA targeting activity in vivo. These results show that RNA-programmed genome editing is a facile strategy for introducing site-specific genetic changes in human cells.

## Introduction

Methods for introducing site-specific double-strand DNA (dsDNA) breaks (DSBs) in genomic DNA have transformed our ability to engineer eukaryotic organisms by initiating DNA repair pathways that lead to targeted genetic re-programming. Zinc-finger nucleases (ZFNs) and transcription activator-like effector nucleases (TALENs) have proved effective for such genomic manipulation but their use has been limited by the need to engineer a specific protein for each dsDNA target site and by off-target activity (*Urnov et al., 2010*; *Bogdanove and Voytas, 2011*). Thus, alternative strategies for triggering site-specific DNA cleavage in eukaryotic cells are of great interest.

Research into genome defense mechanisms in bacteria showed that CRISPR (Clustered Regularly Interspaced Short Palindromic Repeats)/Cas (CRISPR-associated) loci encode RNA-guided adaptive immune systems that can destroy foreign DNA (*Bhaya et al., 2011*; *Terns and Terns, 2011*; *Wiedenheft et al., 2012*). The Type II CRISPR/Cas systems require a single protein, Cas9, to catalyze DNA cleavage (*Sapranauskas et al., 2011*). Cas9 generates blunt DSBs at sites defined by a 20-nucleotide guide sequence contained within an associated CRISPR RNA (crRNA) transcript (*Gasiunas et al., 2012*; *Jinek et al., 2012*). Cas9 requires both the guide crRNA and a trans-activating crRNA (tracrRNA) that is partially complementary to the crRNA for site-specific DNA recognition and cleavage (*Deltcheva et al., 2011*; *Jinek et al., 2012*). Recent experiments showed that the crRNA:tracrRNA complex can be redesigned as a single transcript (single-guide RNA or sgRNA) encompassing the features required for both Cas9 binding and DNA target siterecognition (*Jinek et al., 2012*). Using sgRNA, Cas9 can be programmed to cleave double-stranded DNA at any site defined by the guide RNA sequence and including a GG protospacer-adjacent (PAM) motif (*Sapranauskas et al., 2011*; *Jinek et al., 2012*). These findings suggested the exciting possibility that Cas9:sgRNA complexes might constitute a simple and versatile RNA-directed system for generating DSBs that could facilitate site-specific

**\*For correspondence:** doudna@berkeley.edu

**eLife digest** The ability to make specific changes to DNA—such as changing, inserting or deleting sequences that encode proteins—allows researchers to engineer cells, tissues and organisms for therapeutic and practical applications. Until now, such genome engineering has required the design and production of proteins with the ability to recognize a specific DNA sequence. The bacterial protein, Cas9, has the potential to enable a simpler approach to genome engineering because it is a DNA-cleaving enzyme that can be programmed with short RNA molecules to recognize specific DNA sequences, thus dispensing with the need to engineer a new protein for each new DNA target sequence.

Now Jinek et al. demonstrate the capability of RNA-programmed Cas9 to introduce targeted double-strand breaks into human chromosomal DNA, thereby inducing site-specific genome editing reactions. Cas9 assembles with engineered single-guide RNAs in human cells and the resulting Cas9-RNA complex can induce the formation of double-strand breaks in genomic DNA at a site complementary to the guide RNA sequence. Experiments using extracts from transfected cells show that RNA expression and/or assembly into Cas9 is the limiting factor for the DNA cleavage, and that extension of the RNA sequence at the 3′ end enhances DNA targeting activity in vivo.

These results show that RNA-programmed genome editing is a straightforward strategy for introducing site-specific genetic changes in human cells, and the ease with which it can programmed means that it is likely to become competitive with existing approaches based on zinc finger nucleases and transcription activator-like effector nucleases, and could lead to a new generation of experiments in the field of genome engineering for humans and other species with complex genomes.

genome editing. However, it was not known whether such a bacterial system would function in eukaryotic cells.

We show here that Cas9 can be expressed and localized to the nucleus of human cells, and that it assembles with sgRNA in vivo. These complexes can generate double stranded breaks and stimulate non-homologous end joining (NHEJ) repair in genomic DNA at a site complementary to the sgRNA sequence, an activity that requires both Cas9 and the sgRNA. Extension of the RNA sequence at its 3′ end enhances DNA targeting activity in vivo. Further, experiments using extracts from transfected cells show that sgRNA assembly into Cas9 is the limiting factor for Cas9-mediated DNA cleavage. These results demonstrate the feasibility of RNA-programmed genome editing in human cells.

## Results

To test whether Cas9 could be programmed to cleave genomic DNA in vivo, we co-expressed Cas9 together with an sgRNA designed to target the human clathrin light chain (CLTA) gene. The CLTA genomic locus has previously been targeted and edited using ZFNs (*Doyon et al., 2011*). We first tested the expression of a human-codon-optimized version of the *Streptococcus pyogenes* Cas9 protein and sgRNA in human HEK293T cells. The 160 kDa Cas9 protein was expressed as a fusion protein bearing an HA epitope, a nuclear localization signal (NLS), and green fluorescent protein (GFP) attached to the C-terminus of Cas9 (*Figure 1A*). Analysis of cells transfected with a vector encoding the GFP-fused Cas9 revealed abundant Cas9 expression and nuclear localization (*Figure 1B*). Western blotting confirmed that the Cas9 protein is expressed largely intact in extracts from these cells (*Figure 1A*). To program Cas9, we expressed sgRNA bearing a 5′-terminal 20-nucleotide sequence complementary to the target DNA sequence, and a 42-nucleotide 3′-terminal stem loop structure required for Cas9 binding (*Figure 1C*). This 3′-terminal sequence corresponds to the minimal stem-loop structure that has previously been used to program Cas9 in vitro (*Jinek et al., 2012*). The expression of this sgRNA was driven by the human U6 (RNA polymerase III) promoter (*Medina and Joshi, 1999*). Northern blotting analysis of RNA extracted from cells transfected with the U6 promoter-driven sgRNA plasmid expression vector showed that the sgRNA is indeed expressed, and that their stability is enhanced by the presence of Cas9 (*Figure 1D*).

Next, we investigated whether site-specific DSBs are generated in HEK293T cells transfected with Cas9-HA-NLS-mCherry and the CLTA1 sgRNA. To do this, we probed for minor insertions and deletions in the locus resulting from imperfect repair by DSB-induced NHEJ using the Surveyor nuclease

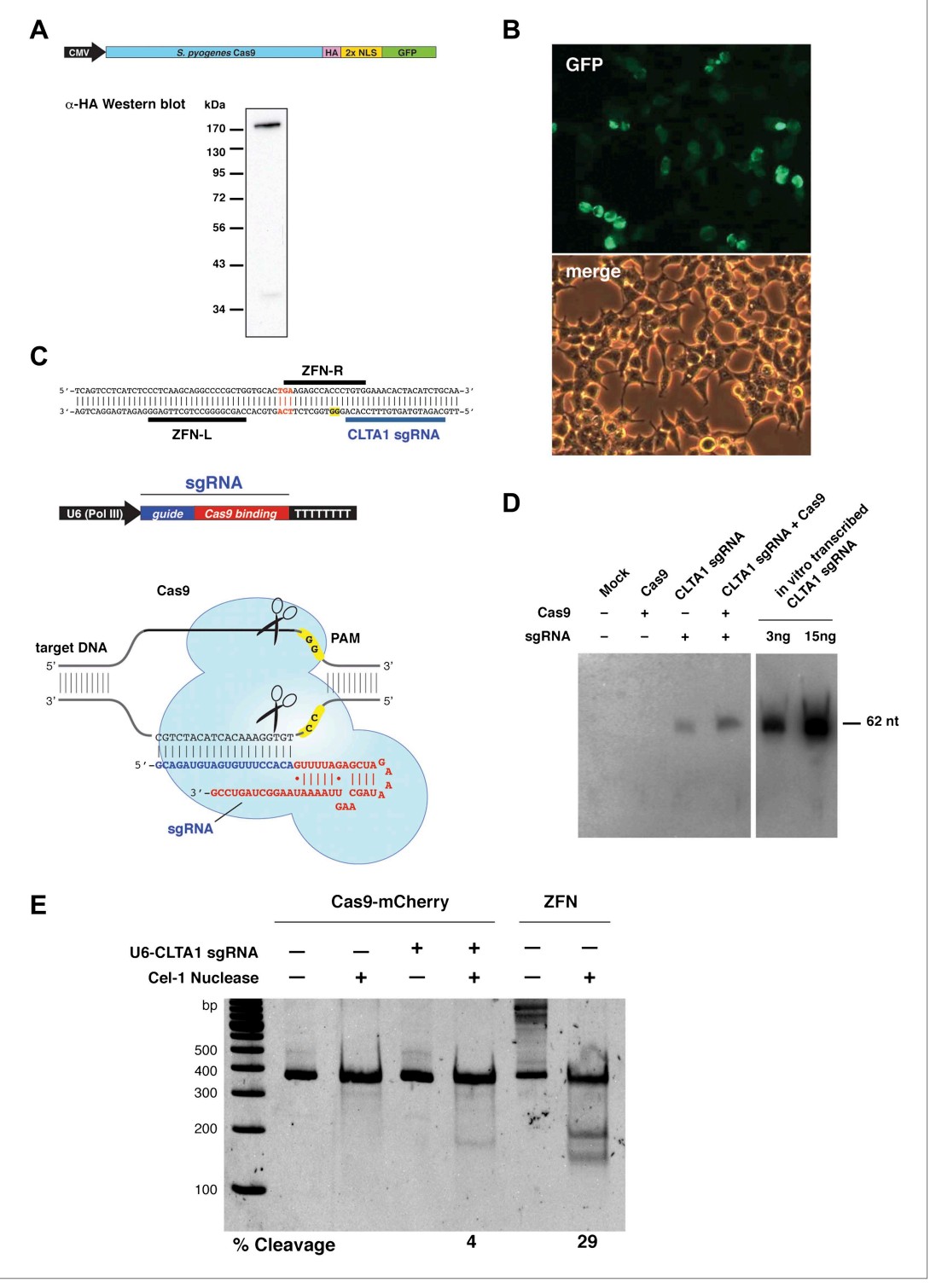

**Figure 1**. Co-expression of Cas9 and guide RNA in human cells generates double-strand DNA breaks at the target locus. (**A**) Top: schematic diagram of the Cas9-HA-NLS-GFP expression construct. Bottom: lysate from HEK293T cells transfected with the Cas9 expression plasmid was analyzed by Western blotting using an anti-HA antibody. (**B**) Fluorescence microscopy of HEK293T cells expressing Cas9-HA-NLS-GFP. (**C**) Blue line indicates the sequence used for the guide segment of CLTA1 sgRNA. Top: schematic diagram of the sgRNA target site in exon 7 of the human CLTA gene. The target sequence that hybridizes to the guide segment of CLTA1 sgRNA is indicated by the blue line. The GG nucleotide protospacer adjacent motif (PAM) is highlighted in yellow. Black lines denote the DNA binding regions

*Figure 1. Continued on next page*

*Figure 1. Continued*

of the control ZFN protein. The translation stop codon of the CLTA open reading frame is highlighted in red for reference. Middle: schematic diagram of the sgRNA expression construct. The RNA is expressed under the control of the U6 Pol III promoter and a poly(T) tract that serves as a Pol III transcriptional terminator signal. Bottom: sgRNA-guided cleavage of target DNA by Cas9. The sgRNA consists of a 20-nt 5′-terminal guide segment (blue) followed by a 42-nt stem-loop structure required for Cas9 binding (red). Cas9-mediated cleavage of the two target DNA strands occurs upon unwinding of the target DNA and formation of a duplex between the guide segment of the sgRNA and the target DNA. This is dependent on the presence of a GG dinucleotide PAM downstream of the target sequence in the target DNA. Note that the target sequence is inverted relative to the upper diagram. (**D**) Northern blot analysis of sgRNA expression in HEK239T cells. (**E**) Surveyor nuclease assay of genomic DNA isolated from HEK293T cells expressing Cas9 and/or CLTA sgRNA. A ZFN construct previously used to target the CLTA locus (**Doyon et al., 2011**) was used as a positive control for detecting DSB-induced DNA repair by non-homologous end joining.

The following figure supplements are available for figure 1:

**Figure supplement 1**. Mutated alleles of the CLTA gene in HEK293T cells as a result of Cas9-induced NHEJ.

assay (**Qiu et al., 2004**). The region of genomic DNA targeted by Cas9:sgRNA is amplified by PCR and the resulting products are denatured and reannealed. The rehybridized PCR products are incubated with the mismatch recognition endonuclease Cel-1 and resolved on an acrylamide gel to identify Cel-1 cleavage bands. As DNA repair by NHEJ is typically induced by a DSB, a positive signal in the Surveyor assay indicates that genomic DNA cleavage has occurred. Using this assay, we detected cleavage of the CLTA locus at a position targeted by the CLTA1 sgRNA (**Figure 1E**). A pair of ZFNs that target a neighboring site in the CLTA locus provided a positive control in these experiments (**Doyon et al., 2011**).

To determine if either Cas9 or sgRNA expression is a limiting factor in the observed genome editing reactions, lysates prepared from the transfected cells were incubated with plasmid DNA harboring a fragment of the CLTA gene targeted by the CLTA1 sgRNA. Plasmid DNA cleavage was not observed upon incubation with lysate prepared from cells transfected with the Cas9-HA-NLS-GFP expression vector alone, consistent with the Surveyor assay results. However, robust plasmid cleavage was detected when the lysate was supplemented with in vitro transcribed CLTA1 sgRNA (**Figure 2A**). Furthermore, lysate prepared from cells transfected with both Cas9 and sgRNA expression vectors supported plasmid cleavage, while lysates from cells transfected with the sgRNA-encoding vector alone did not (**Figure 2A**). These results suggest that a limiting factor for Cas9 function in human cells could be assembly with the sgRNA. We tested this possibility directly by analyzing plasmid cleavage in lysates from cells transfected as before in the presence and absence of added exogenous sgRNA. Notably, when exogenous sgRNA was added to lysate from cells transfected with both the Cas9 and sgRNA expression vectors, a substantial increase in DNA cleavage activity was observed (**Figure 2B**). This result indicates that the limiting factor for Cas9 function in HEK293T cells is the expression of the sgRNA or its loading into Cas9.

As a means of enhancing the Cas9:sgRNA assembly in vivo, we next tested the effect of extending the presumed Cas9-binding region of the guide RNA. Two new versions of the CLTA1 sgRNA were designed to include an additional 4 or 10 base pairs in the helix that mimics the base-pairing interactions between the crRNA and tracrRNA (**Figure 3A**). Additionally, the 3′-end of the guide RNA was extended by five nucleotides based on the native sequence of the *S. pyogenes* tracrRNA (**Deltcheva et al., 2011**). Vectors encoding these 3′ extended sgRNAs under the control of either the U6 or H1 Pol III promoters were transfected into cells along with the Cas9-HA-NLS-GFP expression vector and site-specific genome cleavage was tested using the Surveyor assay (**Figure 3B**). These results suggest that the 3′-extended sgRNAs support more efficient Cas9 function in vivo, although more quantitative comparisons will be necessary to confirm this conclusion.

## Discussion

Besides serving as an invaluable research tool, targeted genome engineering in cells and organisms could potentially provide the path to revolutionary applications in human therapies, agricultural biotechnology and microbial engineering. Methods of modifying the genome exploit endogenous DNA repair pathways that are initiated by the introduction of site-specific dsDNA cleavages. The results presented here provide a straightforward system of RNA-guided site-specific dsDNA cleavage using

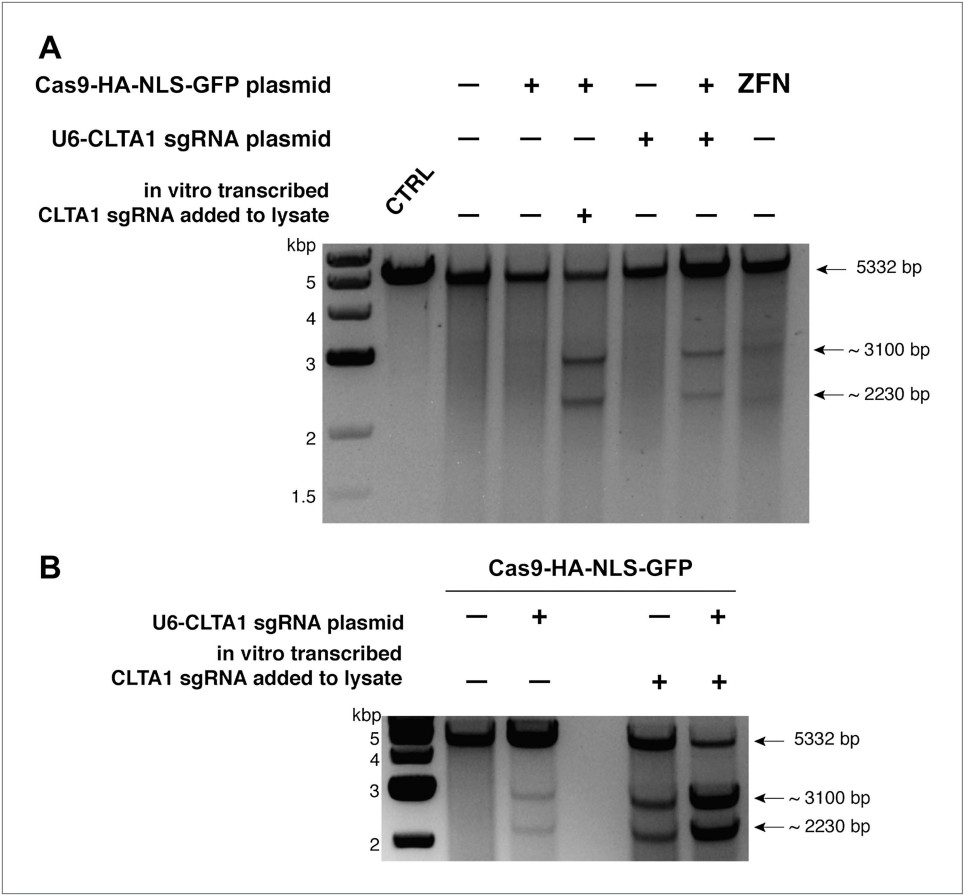

**Figure 2**. Cell lysates contain active Cas9:sgRNA and support site-specific DNA cleavage. (**A**) Lysates from cells transfected with the plasmid(s) indicated at left were incubated with plasmid DNA containing a PAM and the target sequence complementary to the CLTA1 sgRNA; where indicated, the reaction was supplemented with 10 pmol of in vitro transcribed CLTA1 sgRNA; secondary cleavage with XhoI generated fragments of ~2230 and ~3100 bp fragments indicative of Cas9-mediated cleavage. A control reaction using lysate from cells transfected with a ZFN expression construct shows fragments of slightly different size reflecting the offset of the ZFN target site relative to the CLTA1 target site. (**B**) Lysates from cells transfected with Cas9-HA-NLS-GFP expression plasmid and, where indicated, the CLTA1 sgRNA expression plasmid, were incubated with target plasmid DNA as in (**A**) in the absence or presence of in vitro-transcribed CLTA1 sgRNA.

the Cas9 protein from a Type II bacterial CRISPR system to promote genome editing in human cells. Our data show that a codon-optimized version of Cas9, when programmed by an appropriate sgRNA, successfully assembles into Cas9 targeting complexes to trigger site-specific DNA cleavage and repair by NHEJ. The efficiency of NHEJ-induced mutagenesis at the CLTA locus investigated here is consistently in the range of 6–8%. This frequency is lower than that found for a ZFN pair that recognizes a nearby target sequence, but is within the range of frequencies observed more generally with ZFNs and TALENs (**Bogdanove and Voytas, 2011**). Our data suggest that sgRNA expression and/or its assembly into Cas9, rather than Cas9 expression, localization or folding, presently limits Cas9 function in human cells. Higher efficiencies of Cas9-mediated genome targeting could be achieved by optimization of the sgRNA construct design, its expression levels or its subcellular localization. We note that the sgRNAs used in this study are not thought to be 5'-capped or 3'-polyadenylated, which may have reduced their stability in vivo. This and other 5' and 3' end modifications might provide alternative approaches to enhancing Cas9:sgRNA assembly and activity in cells. Nonetheless, the levels of targeting observed in this study have been obtained with a minimal system that relies on simple base pairing to a guide RNA, in contrast to the ZFN and TALEN proteins, which require a new protein to be

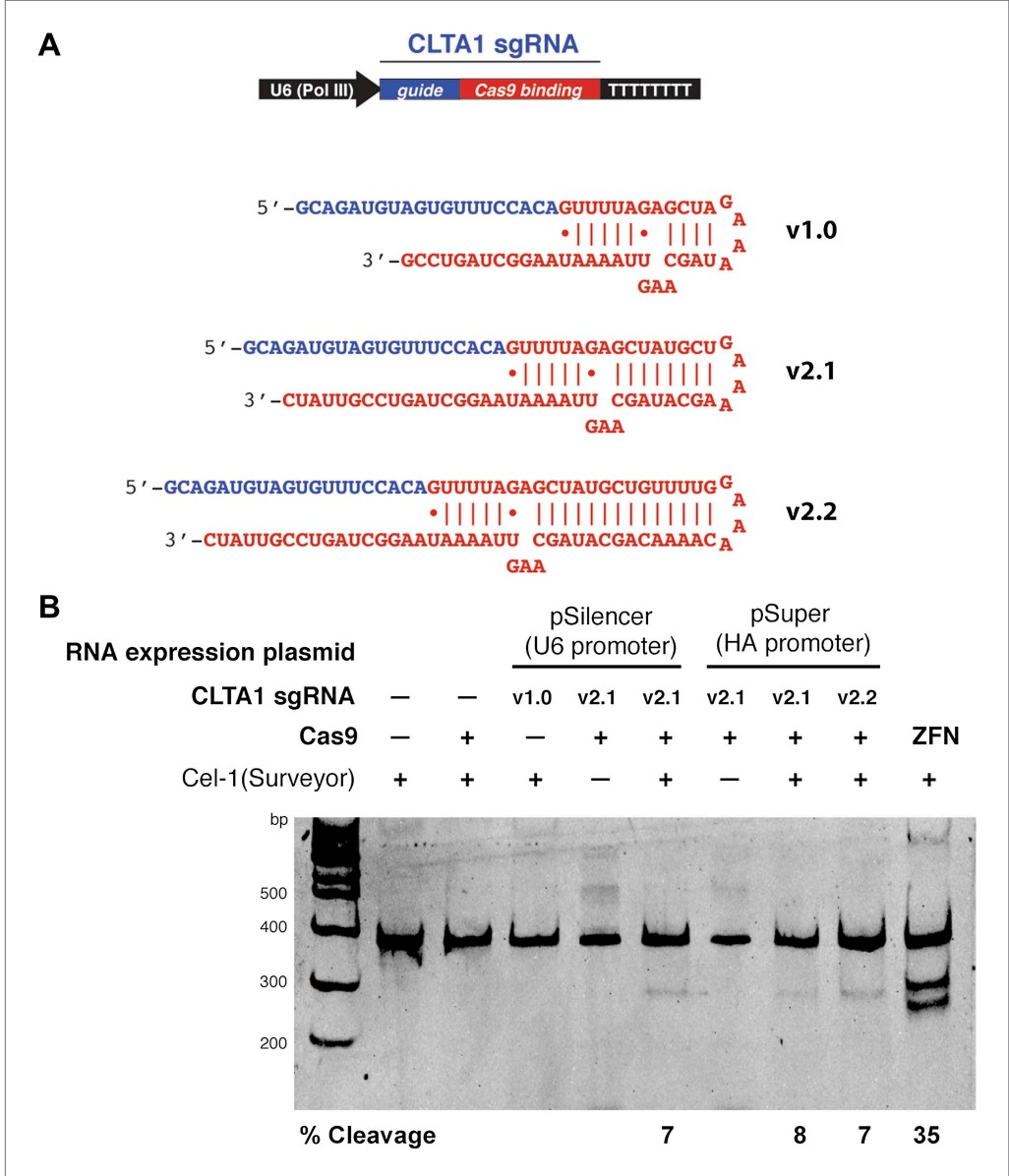

**Figure 3**. 3′ extension of sgRNA constructs enhances site-specific NHEJ-mediated mutagenesis. (**A**) The construct for CLTA1 sgRNA expression (top) was designed to generate transcripts containing the original Cas9-binding sequence v1.0 (***Jinek et al., 2012***), or sequences extended by 4 base pairs (v2.1) or 10 base pairs (v2.2). (**B**) Surveyor nuclease assay of genomic DNA isolated from HEK293T cells expressing Cas9 and/or CLTA sgRNA v1.0, v2.1 or v2.2. A ZFN construct previously used to target the CLTA locus (***Doyon et al., 2011***) was used as a positive control for detecting DSB-induced DNA repair by non-homologous end joining.

engineered for each new cleavage site. RNA-guided genome editing would thus offer distinct advantages due to the simplicity of the sgRNA design.

Our results thus provide the framework for implementing Cas9 as a facile molecular tool for diverse genome editing applications. Although not tested explicitly in this study, a powerful feature of this system is the potential to program Cas9 with multiple sgRNAs in the same cell, either to increase the efficiency of targeting at a single locus, or as a means of targeting several loci simultaneously. Such strategies would find broad application in genome-wide experiments and large-scale research efforts such as the development of multigenic disease models. As an inexpensive and rapid mechanism for triggering site-specific genome modification, the programmable Cas9:sgRNA system could potentially transform next-generation genome-scale studies.

# Materials and methods

## Plasmid design and construction

The sequence encoding *Streptococcus pyogenes* Cas9 (residues 1–1368) fused to an HA epitope (amino acid sequence DAYPYDVPDYASL), a nuclear localization signal (amino acid sequence PKKKRKVEDPKKKRKVD) was codon optimized for human expression and synthesized by GeneArt (Regensburg, Germany; DNA and protein sequences shown in *supplementary file 1*). Ligation-independent cloning (LIC) was used to insert this sequence into a pcDNA3.1-derived GFP and mCherry LIC vectors (vectors 6D and 6B, respectively, obtained from the UC Berkeley MacroLab), resulting in a Cas9-HA-NLS-GFP and Cas9-HA-NLS-mCherry fusions expressed under the control of the CMV promoter. Guide sgRNAs were expressed using expression vector pSilencer 2.1-U6 puro (Life Technologies, Carlsbad, CA) and pSuper (Oligoengine, Seattle, WA). RNA expression constructs were generated by annealing complementary oligonucleotides to form the RNA-coding DNA sequence and ligating the annealed DNA fragment between the BamHI and HindIII sites in pSilencer 2.1-U6 puro and BglII and HindIII sites in pSuper.

## Cell culture conditions and DNA transfections

HEK293T cells were maintained in Dulbecco's modified eagle medium (DMEM) supplemented with 10% fetal bovine serum (FBS) in a 37°C humidified incubator with 5% $CO_2$. Cells were transiently transfected with plasmid DNA using either X-tremeGENE DNA Transfection Reagent (Roche Applied Science, Indianapolis, IN) or Turbofect Transfection Reagent (Thermo Scientific, Waltham, MA) with recommended protocols. Briefly, HEK293T cells were transfected at 60–80% confluency in 6-well plates using 0.5 µg of the Cas9 expression plasmid and 2.0 µg of the RNA expression plasmid. The transfection efficiencies were estimated to be 30–50% for Turbofect (*Figures 1E and 2A,B*) and 80–90% for X-tremegene (*Figure 3B*), based on the fraction of GFP-positive cells observed by fluorescence microscopy. 48 hr post transfection, cells were washed with phosphate buffered saline (PBS) and lysed by applying 250 µl lysis buffer (20 mM Hepes pH 7.5, 100 mM potassium chloride [KCl], 5 mM magnesium chloride [$MgCl_2$], 1 mM dithiothreitol [DTT], 5% glycerol, 0.1% Triton X-100, supplemented with Roche Protease Inhibitor cocktail) and then rocked for 10 min at 4°C. The resulting cell lysate was divided into aliquots for further analysis. Genomic DNA was isolated from 200 µl cell lysate using the DNeasy Blood and Tissue Kit (Qiagen, Hilden, Germany) according to the manufacturer's protocol.

## Western blot analysis of Cas9 expression

HEK293T, transfected with the Cas9-HA-NLS-GFP expression plasmid, were harvested and lysed 48 hr post transfection as above. 5 µl of lysate were electrophoresed on a 10% SDS polyacrylamide gel, blotter onto a PVDF membrane and probed with HRP-conjugated anti-HA antibody (1:1000 dilution in 1× PBS; Sigma, St. Louis, MO).

## Surveyor assay

The Surveyor assay was performed as previously described (*Qiu et al., 2004*; *Miller et al., 2007*; *Doyon et al., 2011*). Briefly, the human clathrin light chain A (CLTA) locus was PCR amplified from 200 ng of genomic DNA using a high fidelity polymerase, Herculase II Fusion DNA Polymerase (Agilent Technologies, Santa Clara, CA) and forward primer 5′-GCAGCAGAAGAAGCCTTTGT-3′ and reverse primer 5′-TTCCTCCTCTCCCTCCTCTC-3′. 300 ng of the 360 bp amplicon was then denatured by heating to 95°C and slowly reannealed using a heat block to randomly rehybridize wild type and mutant DNA strands. Samples were then incubated with Cel-1 nuclease (Surveyor Kit; Transgenomic, Omaha, NE) for 1 hr at 42°C. Cel-1 recognizes and cleaves DNA helices containing mismatches (wild type:mutant hybridization). Cel-1 nuclease digestion products were separated on a 10% acrylamide gel and visualized by staining with SYBR Safe (Life Technologies). Quantification of cleavage bands was performed using ImageLab software (Bio-Rad, Hercules, CA). The percent cleavage was determined by dividing the average intensity of cleavage products (160–200 bps) by the sum of the intensities of the uncleaved PCR product (360 bp) and the cleavage product. Mutations (indels) resulting from Cas9-induced DNA repair reactions were identified by Sanger sequencing of cloned amplicons. Briefly, the 360-bp PCR amplicon was treated with Taq DNA polymerase (NEB) and dATP to attach an 'A' base to the 3′ end of the DNA fragment. The DNA was then ligated into pGEM-T Easy vector (Promega, Madison, WI) according to the manufacturer's protocol. Transformants were grown on Luria Broth agar plates containing 100 µg/ml of ampicillin, 70 µg/ml 5-bromo-4-chloro-indolyl-β-D-galactopyranoside and 100 µM

isopropyl β-D-1-thiogalactopyranoside. Random white colonies were selected for Sanger sequencing (Quintara Biosciences, Albany, CA). Representative DNA sequences are shown in *Figure 1—figure supplement 1*.

### In vitro transcription

Guide RNA was in vitro transcribed using recombinant T7 RNA polymerase and a DNA template generated by annealing complementary synthetic oligonucleotides as previously described (*Sternberg et al., 2012*). RNAs were purified by electrophoresis on 7 M urea denaturing acrylamide gel, ethanol precipitated, and dissolved in DEPC-treated water.

### Northern blot analysis

RNA was purified from HEK293T cells using the mirVana small-RNA isolation kit (Ambion). For each sample, 800 ng of RNA were separated on a 10% urea-PAGE gel after denaturation for 10 min at 70°C in RNA loading buffer (0.5× TBE [pH 7.5], 0.5 mg/ml bromophenol blue, 0.5 mg xylene cyanol and 47% formamide). After electrophoresis at 10 W in 0.5× TBE buffer until the bromophenol blue dye reached the bottom of the gel, samples were electroblotted onto a Nytran membrane at 20 V for 1.5 hr in 0.5× TBE. The transferred RNAs were cross-linked onto the Nytran membrane in UV-Crosslinker (Strategene) and were pre-hybridized at 45°C for 3 hr in a buffer containing 40% formamide, 5× SSC, 3× Dernhardt's solution (0.1% each of ficoll, polyvinylpyrollidone, and BSA) and 200 µg/ml Salmon sperm DNA. The pre-hybridized membranes were incubated overnight in the prehybridization buffer supplemented with 5'-$^{32}$P-labeled antisense DNA oligo probe at 1 million cpm/ml. After several washes in SSC buffer (final wash in 0.2× SCC), the membranes were imaged phosphorimaging.

### In vitro cleavage assay

Cell lysates were prepared as described above and incubated with CLTA-RFP donor plasmid (*Doyon et al., 2011*). Cleavage reactions were carried out in a total volume of 20 µl and contained 10 µl lysate, 2 µl of 5× cleavage buffer (100 mM HEPES pH 7.5, 500 mM KCl, 25 mM MgCl$_2$, 5 mM DTT, 25% glycerol) and 300 ng plasmid. Where indicated, reactions were supplemented with 10 pmol of in vitro transcribed CLTA1 sgRNA. Reactions were incubated at 37°C for 1 hr and subsequently digested with 10 U of XhoI (NEB) for an additional 30 min at 37°C. The reactions were stopped by the addition of Proteinase K (Thermo Scientific) and incubated at 37°C for 15 min. Cleavage products were analyzed by electrophoresis on a 1% agarose gel and stained with SYBR Safe. The presence of ~2230 and ~3100 bp fragments is indicative of Cas9-mediated cleavage.

## Acknowledgements

We thank David Drubin, Barbara Meyer and Te-Wen Lo for helpful discussions and expert advice; Jamie Cate, Andy May and Rachel Haurwitz for comments on the manuscript; George Church for sharing unpublished data; and Kaihong Zhou and Alison Smith for excellent technical support. This work was funded by the Howard Hughes Medical Institute and by National Institutes of Health Grant R01 GM65462 to David Drubin. J.A.D. is a Howard Hughes Medical Institute investigator.

## Additional information

### Competing interests

MJ, Founder of Caribou Biosciences and member of its scientific advisory board. Has filed a provisional patent related to this work. JD, Founder of Caribou Biosciences and member of its scientific advisory board. Has filed a provisional patent related to this work. The other authors declare that no competing interests exist

### Funding

| Funder | Grant reference number | Author |
| --- | --- | --- |
| Howard Hughes Medical Institute | | Jennifer Doudna |
| National Institutes of Health | | Aaron Cheng |

The funders had no role in study design, data collection and interpretation, or the decision to submit the work for publication.

## Author contributions

MJ, Conception and design, Acquisition of data, Analysis and interpretation of data, Drafting or revising the article; AE, Conception and design, Acquisition of data, Analysis and interpretation of data, Drafting or revising the article; AC, Conception and design, Drafting or revising the article; SL, Acquisition of data, Analysis and interpretation of data; EM, Acquisition of data, Drafting or revising the article; JD, Conception and design, Analysis and interpretation of data, Drafting or revising the article

## Additional files

### Supplementary files

• Supplementary file 1. DNA and protein sequences relating to plasmid design and construction.

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
