## [Author Response]

*1. A few of the mutated targets must be sequenced (from cloned PCR products), just to demonstrate that the expected types of NHEJ mutations are present*.

We agree that sequencing data will be useful for determining the nature and range of NHEJ reactions that are induced by Cas9:sgRNA. However, we note that the Surveyor assay is a well-established stand-alone method of detecting the presence of mutations introduced by NHEJ events as a consequence of double-stranded DNA breaks. For example, in a study that demonstrates improvements on the ZFN architecture for increased efficiency of gene editing, Miller et al. (An improved zinc-finger nuclease architecture for highly specific genome editing, *Nature Biotechnology*; PMID 17603475) used solely the Surveyor assay to monitor genome modification. In a study showing improvements on methods to increase efficiency of genome editing, Doyon et al. (Transient cold shock enhances zinc-finger nuclease–mediated gene disruption, *Nature Methods*; PMID 20436476) use only the Surveyor assay to show that subjecting cells to transient hypothermia can increase efficiencies of ZFN-induced gene disruption. A recent study by Miller et al. (A TALE nuclease architecture for efficient genome editing, *Nature Biotechnology*; PMID 21179091), which was the first to demonstrate the use of TALENs for genome editing, relied solely on the Surveyor assay to demonstrate both NHEJ and homologous recombination at the CCR5 genomic locus, which comprised the majority of the article.

We also point out that cloning and sequencing PCR products to demonstrate genome modification would require analyses of ∼100 clones to obtain 6–8 modified alleles at best. This type of analysis will not provide statistically meaningful data. We think that this analysis should be performed using deep-sequencing methods to provide much larger data sets for multiple Cas9:sgRNA-targeted loci. This will be better performed as part of a larger study that provides in-depth analysis subsequent to the publication of this initial discovery.

*2. The authors mention that off-target effects are an important concern for genome engineering methods. We would like to encourage the authors to look at least at the top loci with sequence similarity to the targeting site and perform the Surveyor assay on these, similar to what was done by Meng et al. (Nature Biotechnology 2008), unless this unduly delays publication*.

We agree that assessing off-target effects is an important future goal, but we think that this is more appropriate for a separate, larger-scale study using deep-sequencing to evaluate off-target effects at many loci. It is clear from our previous results that target recognition *in vitro* requires at least 16 base pairs, including the PAM (Jinek et al. (2012) *Science*, PMID 22745249). We also have a manuscript currently in review at another journal showing that, in bacteria, there are *no* detectable off-target effects for Cas9:sgRNA-mediated gene silencing. Whether this holds true for the much larger human genome will need to be thoroughly investigated, but the results to date are encouraging.